# Checkpoint Inhibitors in Acute Myeloid Leukemia

**DOI:** 10.3390/biomedicines11061724

**Published:** 2023-06-15

**Authors:** Daniela Damiani, Mario Tiribelli

**Affiliations:** 1Division of Hematology and Stem Cell Transplantation, Udine Hospital, 33100 Udine, Italy; mario.tiribelli@uniud.it; 2Department of Medicine, Udine University, 33100 Udine, Italy

**Keywords:** acute myeloid leukemia, immune microenvironment, checkpoint inhibitors, drug resistance

## Abstract

The prognosis of acute myeloid leukemia (AML) remains unsatisfactory. Among the reasons for the poor response to therapy and high incidence of relapse, there is tumor cell immune escape, as AML blasts can negatively influence various components of the immune system, mostly weakening T-cells. Since leukemic cells can dysregulate immune checkpoints (ICs), receptor-based signal transductors that lead to the negative regulation of T-cells and, eventually, to immune surveillance escape, the inhibition of ICs is a promising therapeutic strategy and has led to the development of so-called immune checkpoint inhibitors (ICIs). ICIs, in combination with conventional chemotherapy, hypomethylating agents or targeted therapies, are being increasingly tested in cases of AML, but the results reported are often conflicting. Here, we review the main issues concerning the immune system in AML, the main pathways leading to immune escape and the results obtained from clinical trials of ICIs, alone or in combination, in newly diagnosed or relapsed/refractory AML.

## 1. Introduction

Acute myeloid leukemia (AML) is a heterogeneous disease in which conventional treatment is still associated with unacceptably high rates of relapse and death despite complete remission (CR) being obtained in around 70% of patients [1]. The disease results from the transformation of rapidly proliferating progenitor cells. In AML, dysfunctional hierarchy prevails over the normal one, resulting in progressive cytopenias and increased risks of bleeding and infections.

In recent years, much progress has been made concerning the knowledge of the molecular pathogenesis of AML [2,3], and new drugs acting on various genetic abnormalities have been developed, thus permitting novel therapeutic approaches and, eventually, higher survival rates [4]. The treatment strategies used for AML depend on many factors, both disease-specific (e.g., prior exposure to toxics, prior myelodysplastic syndrome—MDS—cytogenetic and molecular abnormalities, and relapsed or refractory disease) and patient-specific (e.g., age, comorbidities, and previous exposure to chemotherapy). The current treatment for most cases of AML, excluding acute promyelocytic leukemia, consists of two phases: CR induction and the consolidation phase. In patients eligible for intensive induction, the “classical” 3 + 7 regimen is generally used; alternatives include the addition of fludarabine (FLAI) or mitoxantrone-based regimens [5]. In addition, the use of the FLT3 inhibitor midostaurin has become a standard in AML patients harboring FLT3-ITD mutation [5,6]. In patients not eligible for intensive chemotherapy, hypomethylating agents (HMAs), alone or in association with the BCL2 inhibitor venetoclax or IDH1 inhibitor ivosidenib, are the most recent options [5]. In “fit” patients, consolidation includes intermediate or high-dose cytarabine and allogeneic hematopoietic cell transplantation (HCT), which is still considered the only curative option [7]. However, the “cure” of AML remains a major challenge, as almost half of transplanted patients relapse after HCT, a risk that depends on disease status at transplant, donor type, T cell depletion strategies, and conditioning regimen [8]. It has now become evident that AML development and progression, as well as drug resistance acquisition, are associated with changes in the bone marrow microenvironment, which provide physical protection and release pro-survival factors for leukemic cells and induce dysregulated immune responses that facilitate the immune escape of AML blasts [9]. In addition to targeting molecular alterations, the manipulation of the AML immune microenvironment to restore immune surveillance appears to be an attractive strategy. In this paper, we will summarize the current knowledge concerning immune cell impairment in leukemic bone marrow niches and the available tools to harness antileukemic immune response by blocking checkpoint interactions between T/NK effectors and AML blasts.

## 2. T-Cells in the Bone Marrow

The bone marrow (BM) microenvironment is constituted by heterogeneous cell populations, blood vessels, and molecules allocated in niches, providing regulatory signals that contribute to physiological homeostasis throughout life. Its activities span from supporting myelopoiesis, controlling myeloid progenitor differentiation and distribution, regulating mature cell production through fine opposite signals, producing immune cells by supporting lymphopoiesis and acting as a sanctuary for mature lymphoid cell types [10]. As a matter of fact, the substantial amount of immune cells makes BM a lymphoid organ that plays an active role in immunity [10]. BM lymphocytes are distributed throughout the stroma and the marrow parenchyma condensed in follicles, without a clear spatially distinction between B-cell and T-cell compartments, with a B/T ratio of 1:5. Plasma cells account for about 1% of mononuclear cells [11]. Most BM T-cells have already encountered an antigen, as proven by the expression of high levels of CD44 and low levels of CD45RA, thus representing a “BM memory reservoir”, but some circulating naïve T-cells are primed in BM [12]. Moreover, the BM microenvironment promotes memory T cell proliferation, supporting their homeostasis [11].

Beyond a specific anti-pathogen immune function, there is evidence of the role of T-cells in supporting hematopoiesis that emerges during fetal development and persists throughout life [13]. It is well known that patients receiving T-cell-depleted allogeneic HCT have a higher risk of graft failure [14], and the role of T-cells in supporting engraftment has been demonstrated via the co-administration of CD3 cells and allogeneic BM-derived stem cells, with positive results [15]. BM T-cells are also involved in the regulation of maturation of hematopoietic precursors and in the release of mature cells. Athymic mice models exhibit peripheral neutropenia and BM immature granulocytic cell accumulation, an effect restored via a thymus graft or through the adoptive transfer of activated (but not naïve) CD4+ cells [16,17]. T helper polarization also contributes to normal hematopoiesis: in murine models, Th2-biased response induced a decrease in the number and cycling status of hematopoietic precursors, while the Th1-biased response shows an increased number of progenitors and cycling cells [18], suggesting the role of age-related thymus involution in AML pathogenesis [19]. As well as CD4+, activated CD8+ T-lymphocytes are involved in the regulation of the number of myeloid multipotent cells and committed precursors. IFNγ, secreted by CD8+ cells, induces mesenchymal cells to release IL6, which modulates the expression of transcription factors crucial for myeloid commitment and promotes the release of mature cells in the peripheral blood [20]. After HCT, memory CD8+ T-cells exert anti-leukemia effects by limiting the potentially dangerous effects of graft-versus-host disease (GvHD) [21,22], confirming that BM CD8+ cells are functionally distinct from those in other compartments and possess higher anti-tumor activity [23]. In addition to effector T-cells, regulatory T-cells (Tregs) are also involved in hematopoiesis. The frequency of FoxP3+ Tregs in the BM is higher (about 20%) than in the spleen and lymph nodes (10%), suggesting that BM is a preferential site of migration or selective function for Tregs [24]. In BM, regulatory cells localize close to the endosteum surface, and, in HCT animal models, Tregs depletion is associated with a 70% reduction in donor cells, suggesting the role of Tregs in favoring engraftment by providing mechanisms to escape graft rejection [25]. Urbieta and colleagues demonstrated that Tregs can prevent CFU progenitor cell differentiation and hypothesized Tregs involvement in maintaining the stem cell pool, although the precise molecular mechanisms are still under investigation [26]. The imbalance between T-helper (Th1/Th2) cells, T-killer (Tc1/Tc22) cells and Th17/Tregs significantly modifies cytokine levels and influences hematopoietic function. A type I immune T-cell response polarization, with high IFN-γ production, as well as an elevated Th17/Treg ratio, is associated with poor graft function after allogeneic HCT [27]; a prevalence of Th1, Tc1, and Th17 cells has been observed in patients with persistent thrombocytopenia after HCT [28]. BM T-cells are also involved in bone remodeling. Normal “bone” density depends on T-cell activation, inducing the secretion of osteoprotegerin by B-cells, thus balancing mineral osteoblast deposition and osteoclast bone reabsorption. Osteoblast activity seems to increase the number of primitive progenitors, and osteoclast activity favors stem cell mobilization [29,30]. Therefore, if T-cells are highly activated, they are turned in an osteoclastogenic phenotype that promotes niche disruption, loss of hematopoietic function, and osteolytic disease [31]. Altogether, these data underline the importance of BM immune cells in mediating immunity, hematopoiesis, and bone modeling and highlight their role in hematopoietic stem cell protection and homeostasis.

## 3. Immune Landscape in AML

Much evidence suggests that the BM microenvironment is remodeled by and for leukemic cells to induce drug resistance and favor immune evasion. From an immunological perspective, the MDS/AML microenvironment results from a complex network of intrinsic and extrinsic factors recruited by neoplastic stem cells to induce immunosuppression and drive disease progression. However, microenvironment changes are peculiar to different myeloid diseases: in low-risk MDS, an inflammatory microenvironment and immunosenescence are prevalent [32,33,34], while in high-risk MDS and AML, the landscape is characterized by clonal expansion and immunosuppression [35,36,37]. In AML, the remodeling of the microenvironment affects all of the components of a successful anti-tumor immune response, inducing low neo-antigen burden and defective antigen presentation, an imbalance between T-effector cells and Tregs, T-cell exhaustion through the upregulation of immune checkpoint receptors and ligands, increased levels of myeloid suppressor cells (MDSCs) and suppressive macrophage (M2), increased the production of immunosuppressive soluble factors, reviewed in Austin et al. [38].

### 3.1. Defective Antigen Presentation and Neo-Antigens in AMLs

In immune response activation, the recognition of leukemic neo-antigens is a critical step and depends on MHC class I (in human HLA-A, -B, -C) and class II (HLA DP, DQ, DR) machineries. These “classical” HLA molecules are expressed on the cell surface, while “non classical” molecules (HLA-DN, DO, DM) play intracellular regulatory roles [39].

Class I HLA molecules are expressed on the majority of cells and present to cytotoxic T-cells self-antigens derived from cytosolic proteins. HLA class II expression is restricted to antigen-presenting cells (APCs) and presents extracellular proteins to CD4+ T-cells. Despite acting as APCs, AML cells do not elicit an effective immune response due to the defective formation of immune synapses and antigen presentation to effector cells [40,41]. Studies using AML samples have shown that the total loss of HLA class I molecules is infrequent [42,43] in contrast with solid tumors, where the loss or mutation of HLA class I is a common mechanism of immune escape from T-cell action [44]. However, the sustained expression of HLA class I molecules can facilitate escape from NK cells as they are ligands for NK inhibitory receptors (KIRs) [38]. The expression of “non classical” HLA molecules has been associated with immune suppression [45]. However, immunosuppressive HLA g molecules may contribute to an immunosuppressive microenvironment through the inhibition of dendritic cells, T-cells and NK cells [46].

Class II HLA molecules have variable expressions in AML. The immunoediting properties of leukemic cells and the epigenetic down-regulation of class II HLA molecules have been observed in the context of allogeneic HCT and have been associated with the evasion of blasts from the graft-versus-leukemia (GvL) effect and post-transplant relapse [46,47,48]. Antohe et al. recently investigated, in a small cohort of AML patients, the expression of HLA-DR, several molecules involved in HLA-II antigen presentation (HLA-DM, CD74 and CLIP) and some checkpoint ligands, finding high levels of HLA-DM and CD74 in all patients, but low HLA-DR expression in only 23% of them. Moreover, they observed that the different expression profiles correlate with distinct ELN risk stratification [49].

Beyond impaired APC ability, there is increasing evidence that antigens resulting from recurrent genetic mutations can modulate the immune microenvironment in AML. NPM1 is the most frequently mutated gene in AML, along with FLT3 and DNMT3A; specifically, the NPM1 mutation occurs in about 30% of de novo AML cases [5] and defines a distinct AML subgroup with a favorable prognosis [50]. Greiner et al. demonstrated that mutant NPM1 induces T-cell response [51]. Forghieri et al. observed a robust T-cell response after CR attainment and an inverse correlation with minimal residual disease (MRD) and relapse [52]. An internal tandem duplication of the juxtamembrane domain of the receptor tyrosine kinase FLT3 (FLT3-ITD) is shown in around 30% of AML patients and confers an adverse prognosis [53]. Patients with FLT3-ITD have an expanded population of dendritic cells (DCs), dendritic cell precursors (pre-DCS) and Tregs [54]. The persistence of pre-DCs in patients in CR after classical chemotherapy is associated with an increased relapse risk [55]. Conversely, treatment with midostaurin induces reductions in Tregs and DCs [56]. Mutations of isocitrate dehydrogenase (IDH) 1 and 2 are found in 9–20% of AML cases; mutations produce high levels of the oncometabolite R-2-hydroxyglutarate (R-2-HG), which is known to block terminal differentiation and induce epigenetic rewriting and genomic instability [57,58]. In AML, R-2-HG induces Tregs [59]. The p53 tumor suppressor is a transcription factor that regulates multiple cell functions, including mitotic division, DNA repair, apoptosis and senescence [60]. The TP53 mutation is present in about 6% of AML patients and is associated with very poor outcomes [61]. P53 mutation in AML activates the transcription of the regulators of innate immune response and NK cell ligands [62,63].

Various evidence recently showed how distinct molecular features, such as NPM1 and FLT3 mutations, differently modulate immune responses in AML, both in terms of immune response quality and immune escape [64,65,66].

### 3.2. Dendritic Cells (DCs), Myeloid-Derived Suppressor Cells (MDSCs) Macrophages (Mas) and Mesenchymal Stromal Cells (MSCs) in AML

Derolf et al. demonstrated that BM DCs are reduced or absent at AML onset and that the attainment of CR was associated with DC regeneration, though no correlation between DCs and survival was found [67]. On the contrary, an elevated proportion of DCs in peripheral blood were associated with an immunosuppressive phenotype [54].

MDSCs physiologically regulate the suppression of T-cells and NK cells. In AML, an increased number of MDSCs has been observed both in BM and in the peripheral blood and correlate with MRD positivity [68]. MDSC proliferation can be stimulated by the same AML cells via extracellular vesicles containing the oncoprotein MUC1, which is able to increase cMYC expression and, ultimately, MDSC growth [68]. MDSC suppression mechanisms include the inhibition of T-cell activation, PD-L1, IDO1, TGFβ, IL-10, and ROS [69].

Higher numbers of M2 macrophages, with anti-inflammatory and tumor-supporting properties, have been shown in AML compared to healthy controls. AML blasts promote the M2 phenotype in macrophages via arginase II production [70]. M2 polarization may be mediated by transcription factor Gfi-1 [71], but also epigenetic regulators and miRNAs may contribute to inhibiting M1 polarization [72]. Yang et al. found that in AML patients, macrophages are heterogenous, with both M1 and M2 subtypes, and that M2 macrophages were associated with poor prognosis [73].

In murine leukemia models, MSCs are higher at AML diagnosis and are reduced in CR [74]. MSCs generate a highly immunosuppressive microenvironment through the induction of Tregs and the upregulation of IDO1 [75]. Moreover, they protect AML blast cells from NK cell killing and inhibit NK expansion [76,77].

### 3.3. Natural Killer Cells in AML

In AML, NK cells are reduced in number and have impaired function. Function inhibition is mediated by Tregs via TGF-β [78]. Previous data associate the function and frequency of NK cells with poor prognosis [79]. In patients achieving CR but not in non-responders, the functional and phenotypic dysfunctions of NK were reversed [80]. NK cell dysfunction, resulting in their inability to lyse leukemic cells, is a consequence of the poor expression of natural cytotoxic receptors [38,81], the reduced production of cytotoxic cytokines [82], and the downregulation of activating receptors [38,83]. On the other hand, the enhanced expression of inhibitory KIR receptors has been observed in AML patients [84]. Finally, NK cells show the upregulation of co-inhibitory receptors, such as PD1, Tim-3 and TIGIT, which correlate with poor function and resemble exhausted T cell status [85].

### 3.4. Tregs in AML

There is consensus among studies on the number and function of Tregs cells in AML patients, as most documented a higher number and increased function of Tregs in the peripheral blood of AML patients [86,87,88,89,90]. Shenghui et al. also reported higher Tregs in the BM (11.9 vs. 9.19%, *p* < 0.001) and observed more immunosuppressive activity in BM resident Tregs compared to peripheral blood ones, suggesting a preferential accumulation of Tregs in the BM and supporting the theory of BM as a privileged niche [88]. Wan et al. recently confirmed that AML Tregs have a higher pro-apoptotic ability on CD4+/CD25− T-cells compared to “normal” Tregs and a higher suppressive activity on CD4+/CD25− IFN-γ secretion compared to that of healthy controls. In addition, they reported that AML Tregs have increased migratory capacity due to an increased expression of CXCR4 when compared to normal Tregs and increased BM. 

B regulatory cells (Bregs), able to convert CD4+/CD25− T-cells in CD4+/CD25+/Foxp3+ Tregs, are comparable in AML and in normal settings, suggesting that different mechanisms are involved in generating the high number of regulatory cells in AML [91]. Tregs numbers vary during treatment. Lichtenegger et al. observed reduced Tregs after CR achievement, but numbers increased during chemotherapy maintenance [92]. Ersvaer et al. showed a reduction after treatment, but absolute numbers were still higher than in normal controls [93]. Many studies hypothesized a prognostic value of Tregs [86,88], suggesting that AML cells utilize Tregs to suppress the immune response and that Tregs recruitment may promote disease persistence [94]. Furthermore, Kanakry et al. found that Tregs represent a dominant population in early lymphocyte recovery after induction chemotherapy [87]. Zeidner et al. reported that the addition of pomalidomide at early lymphocyte recovery after chemotherapy has an immunomodulatory effect, suggesting that early lymphocyte recovery may be the ideal timepoint for immunologic interventions [95]. The expression of inhibitory molecules, such as CTLA-4, PD-1, LAG-3, Tim-3 and TIGIT, is crucial for the suppressive activity [96], making Tregs suppression via checkpoint inhibition a possible approach to promote an efficient anti-tumor immune response.

### 3.5. Effector T-Cells in AML

Scarce and conflicting data are currently available on the frequency and distribution of effector T-cells in AML at diagnosis. Le Dieu et al. observed comparable BM T-cells frequencies among AML patients and healthy subjects but a significant increase in absolute T-cell number in previously untreated AML [41]. Furthermore, in AMLs, they reported a reduced CD4:CD8 ratio due to the expansion of CD3+/CD56+ T-cells. The authors suggested that this may represent a specific local proliferation of T-cells in response to environmental signals secondary to blast cell proliferation or that it could be the consequence of BM T-cell redistribution under the pressure of highly proliferating blast cells [41] and that this increase reverted after chemotherapy [97]. A recent study by Ferraro et al. has shown that immunological phenotypes of AML patients at diagnosis are an important determinant of outcomes to standard-of-care therapies. This study highlights that patients with better outcomes to standard-of-care therapies have more CD4+ T-cells in their BM, intact response to T-cell stimulation and “cold” infiltrates [98]. Similar results have previously been reported by Lamble et al., which demonstrated that T-cell infiltration correlates with AML prognosis [99]. Le Dieu et al. also demonstrated an increase in both naïve and memory CD3+/CD56+ cell subsets and found an increased expression of activation markers (CD69, CD71, CD57, and HLA-DR) via gene expression profile and flow cytometry analyses [41]. Other authors found no differences in circulating T-cells from leukemic and healthy subjects [100].

There is no consensus on T-cell functional status in AML. Schnorfeil et al. reported that T-cell proliferation and cytokine production upon CD28 co-stimulation is not impaired at AML diagnosis [101]. Kornblau et al. reported reduced IFN-γ production in CD4+ T-cells of patients with untreated AML compared to healthy controls and that the cytokine expression is an independent prognostic factor in AML and MDS [102]. Le Dieu et al. observed differences in the T-cell expression of genes involved in cytoskeleton formation and demonstrated, via in vitro studies, that this correlates with an impaired ability to form immune synapses crucial for T-cell activation [41]. It is likely that treatment and disease phase influence T-cell function, and more extensive and comparable studies are needed to definitively clarify T-cell function in AML patients. Among the causes of T-cell dysfunction in AML, a pivotal role is played by the frequent expression in blasts of co-inhibitory ligands, such as Gal-9, PD-L1, and CD155 [103], that, by binding their receptors on T-cells, lead to T-cell exhaustion, with the defective development of memory T-cells and T-cell deletion in the AML microenvironment [104], and, ultimately, disease progression [105]. Williams et al. recently investigated the distribution of T-cell subsets and the expression of checkpoint receptors and ligands in newly diagnosed and relapsed AML. They reported higher expression of inhibitory receptors in BM T-cells of AML patients at diagnosis compared to healthy controls and the co-expression of different inhibitory molecules in relapsed patients; moreover, they observed the expression of immune-checkpoint ligands on the blasts of patients with adverse cytogenetics and p53 mutations [106].

## 4. Checkpoint Inhibitors in AML Therapy

The evidence that immune checkpoint interactions contribute to AML immune evasion, not only at relapse but also in newly diagnosed disease, represents the rationale to use this therapeutic approach to reactivate immune sensitivity through a block of co-inhibitory ligands. From the approval of the first antibody blocking an immune checkpoint in 2011, immune checkpoint inhibitors (ICI) revolutionized the oncology field, resulting in durable response rates in patients with historically limited therapeutic options [107]. After approval in solid malignancies, various molecules, such as ipilimumab, pembrolizumab, cemiplimab, avelumab, and durvalumab, acting on specific targets, are undergoing clinical trials in relation to AML [108]. Due to their limited activity when used as a monotherapy, despite the demonstrated ability of co-inhibitory blockade and improved immune response, combination trials are ongoing. Despite the promising activity of checkpoint inhibitors in solid tumors [109], so far, clinical trials relating to AML have reported controversial results, possibly due to the low mutational burden and the DNA mismatch repair proficiency in AML when compared to solid tumors [110,111]. However, about fifty trials are currently ongoing testing ICI in AML and MDS, either as monotherapy or in combination with chemotherapy or HMAs. Key ICI potential targets in AML are shown in Figure 1.

### 4.1. The CTLA-4/B7 Axis

CTLA-4 (CD152) is a member of the immunoglobulin-related receptors interacting with CD80 and CD86 ligands to deliver an inhibitory signal aimed at terminating immune responses. Moreover, it has a role in Tregs induction and, consequently, in regulating tolerance and autoimmunity [112]. The aberrant expression of CTLA-4 has been reported in AML, with a detrimental effect on disease outcomes [113]. In animal models, blocking CTLA-4 enhanced T-cell activity and suppressed Tregs [114]. In clinical trials relating to melanoma, anti-CTLA-4 ipilimumab increased the Teff/Tregs ratio, enhanced NK activity and restored T-effector function, ultimately prolonging survival [115,116]. Data on AML cell lines and preliminary results from clinical trials confirmed what has been reported in solid tumors [117]. Zhong et al. have shown that anti-CTLA-4 can improve AML-specific T-cell function in terms of frequency, cytotoxic capacity and IFN-γ secretion [118]. Data on the “in vivo” efficacy of ipilimumab are scarce. Davids and coll. reported rates of CR and partial response (PR) of 23% and 9%, respectively, in hematological malignancies receiving single-agent ipilimumab for relapse after HCT [119]. Bashey et al. observed no responses in AML patients among a small cohort of 29 patients with various hematologic malignancies relapsed after HCT [120]. A phase I trial of ipilimumab and a definite dose of Treg-depleted donor lymphocyte infusion (DLI) for AML and MDS relapsed after HCT is recruited (NCT03912064).

HMAs have been shown to upregulate checkpoint receptors and their ligands in AML/MDS models [121]. The “in vitro” data correlates to a clinical study in AML, showing that the azacytidine (AZA)-induced demethylation of PD-1 promoter translates into the increased expression of PD-1 and an unfavorable outcome in the absence of immunotherapy [122]. A phase I trial combining ipilimumab and decitabine (DAC) in relapsed/refractory AML and MDS after allogeneic HCT or HCT-naïve is currently active but not recruiting (NCT2890329).

### 4.2. The PD1/PD-L1 Axis

Programmed death receptor-1 (PD1 or CD279) is a type I transmembrane protein expressed mostly in activated immune cells [105]. PD1 binds two ligands: PD-L1 (CD274) and PD-L2 (CD273). PD-L1 is a member of the B7 family of co-stimulatory/co-inhibitory molecules present in hematopoietic cells and is upregulated or aberrantly expressed in many tumors [105,123]. In AML, the up-regulation of PD1 was reported in T-effector cells and in Tregs [106]; it is hypothesized that PD1 overexpression on CD8+ T-cells may account for CTL dysfunction and immune suppression during AML progression [124]. Moreover, PD1 and T-cell immunoglobulin and mucin domain 3 (TIM-3) co-expression on T-cells has been correlated to T-cell exhaustion in human and murine models [125] and can predict leukemia relapse post-HCT [126]. In addition, the co-expression of PD1 and TIM-3 seems to be more frequent in patients experiencing multiple relapses [106]. On the other hand, the up-regulation of PD-L1 and PD-L2 at diagnosis, relapse and during treatment correlates with resistance to therapy and poor prognosis [127,128,129,130]. PD1/PD-L1 engagement drives an inhibitory signal causing T-cell exhaustion and favoring the immune escape of neoplastic cells [105]. Moreover, it induces the apoptosis of tumor-specific cells and favors Tregs differentiation and resistance to CD8+ mediated cytolysis [131].

Three PD1-blocking antibodies (nivolumab, pembrolizumab and cemiplimab) and three PD-L1 inhibitors (atezolizumab, avelumab and durvalumab) have been approved to date by the FDA for various solid tumors [108]. However, PD1 and PD-L1 inhibitors have still not been approved for AML.

The results of a few clinical trials of drugs against the PD1–PD-L1 axis that have been completed or are ongoing in relation to AML are summarized in Table 1. Nivolumab was used alone as maintenance in high-risk AML [132] or to eliminate MRD. An ongoing phase II trial is exploring the tolerability of nivolumab in relapsed AML or MRD-positive AML after HCT (NCT03146468), while a phase I/Ib was terminated due to early toxicity. Additionally, pembrolizumab has been evaluated as a maintenance therapy in elderly AML patients or as a salvage therapy after relapse post-allogeneic transplant.

Many clinical trials combining epigenetic therapies with ICI are currently recruiting (NCT03969446, NCT02996474, NCT4277442) [133]. However, patients who received prior HMA therapy displayed lower ORR compared to HMA-naïve patients (22% vs. 58%), suggesting that the combination of HMA and ICI may have better efficacy when administered early in the course of the disease. Daver et al. presented the interim results of the NCT02397720 trial, in which patients with relapsed/refractory AML or elderly patients with newly diagnosed AML were parallel assigned to receive AZA + nivolumab or AZA + nivolumab + ipilimumab: in the latter cohort, ORR and CR/PR rates were 44% and 36%, respectively. The combination of AZA and pembrolizumab in relapsed/refractory AML and in newly diagnosed AML patients over 65 years was investigated in the phase II trial NCT02845297, showing modest clinical activity [134].

Another phase II trial of nivolumab, cytarabine and idarubicin in newly diagnosed AML and MDS (NCT02464657) reported that responders had no evidence of MRD at the time of response, and non-responders were more likely to have TP 53 mutations and a higher frequency of BM CD4+ cells co-expressing PD1 and TIM-3 [135]. The phase II NCT02768792 trial is evaluating the efficacy of the association of pembrolizumab and high-dose cytarabine (HiDAC) in relapsed/refractory AML. The preliminary results showed that pretreatment CD8+ TCR diversity was associated with response, as well as increased chemokine receptors in peripheral blood CD8+ cells and the increased expression of genes involved in p53, IFN-γ and Il-6 pathways [136]. No trial is currently ongoing to test cemiplimab in AMLs.

The preliminary results of the combination of AZA and durvalumab in MDS and AML, which are unfit for intensive chemotherapy, showed an ORR of 61.9% in MDS and of 31% in AML and median OS 11.6 months in MDS and 13.0 months in AML (NCT02775903) [137]. Many phases I and II trials investigating the PD-L1 inhibitors avelumab or atezolizumab in association with HMAs (AZA, DAC or guadecitabine) are recruiting (NCT02953561, NCT03395973, NCT02935361, NCT02892318, NCT3154827), but no results are available at present.

**Table 1 biomedicines-11-01724-t001:** Summary of results of a few clinical trials of drugs against PD1–PD-L1 axis completed or ongoing in AML.

Trial Identifier	Drug	Disease	Response	Survival	Ref.
NCT02532231	Nivolumab	HR AML	CR 79% at 6 months, 71% at 12 months	OS 86% at 12 mo., 67% at 18 mo.	[132]
NCT02397720	Nivolumab + AZA	RR AML	ORR 58%, CR/CRi 21%, HI 7%	Median OS 9.2 mo.	[133]
		AZA-naïve AML	ORR 22%		
NCT02464657	Nivolumab + cytarabine + idarubicin	Newly diagnosed AML and MDS	CR/CRi 80%	Median OS 18.5 mo.	[135]
NCT02845297	Pembrolizumab + AZA	RR AML	ORR 32%, CR/CRi 14%	Median OS 10.8 mo.	[134]
		Newly diagnosed elderly AML	CR/CRi 47%	Median OS 13.1 mo.	
NCT02768792	Pembrolizumab + cytarabine	RR AML	ORR 46%, CR/CRi 38%		[136]
NCT02775903	Durvalumab + AZA	Unfit MDS and AML	ORR 61.9% in MDS ORR 31% in AML	Median OS 11.6 mo. MDS Median OS 13 mo. AML	[137]

Legend. AML: acute myeloid leukemia; AZA: azacytidine; CR: complete remission; CRi: complete remission with incomplete recovery; HI: hematologic improvement; ORR: overall response rate; HR; high risk; MDS: myelodysplastic syndrome; OS: overall survival; RR: relapsed/refractory.

### 4.3. The TIM-3/Galectin9 Axis

TIM-3 is a co-inhibitory receptor expressed on CD4+ Th1 cells, CD8+ cytotoxic T-cells (CTLs) and other cells of innate immunity, such as dendritic cells, monocytes, macrophages, mast cells and NK cells, as well as on different neoplastic cells [138,139,140]. TIM-3 gene is coded on chromosome 5q33.2, in the same region of IL4 and IL-5 genes. TIM-3 is a single transmembrane (TM) molecule whose extracellular tail contains an N-terminal IgV domain, followed by a mucin domain with glycosylation sites. After this, there is the TM domain and the cytoplasmic tail in the C-terminus. TIM-3 does not contain classic inhibitory tyrosine-based motifs; however, there is a conserved region with five tyrosine residues that are phosphorylated after the interaction of TIM-3 with their ligands [141]. Among the four known ligands of TIM-3, the first and most studied is galectin-9 (gal-9), which induces the apoptosis of Th1 cells [104], playing a crucial role in tumor cell immune evasion. TIM-3 overexpression in human and murine tumor models results in T-cell dysfunction [142]. TIM-3 is often co-expressed with PD-1, and blocking TIM-3 alone or with other co-inhibitory molecules reverse T-cell exhaustion [130]. In AML, high levels of TIM-3 have been found in immune cells, particularly T-cells and NK cells, promoting immune exhaustion, and on LSCs, where it represents a distinctive marker. The overexpression of TIM-3 has been described in LSCs but not in healthy hematopoietic stem cells [143,144]. Kikushige et al. suggested that, on LSCs, TIM-3 and its ligand create an autocrine loop, leading to the phosphorylation of ERK and AKT. This process results in the induction of the β-catenin pathway and NF-kB activation, which regulate development and self-survival [145]. TIM-3 is involved in immune evasion through different mechanisms. Folgiero et al. established that the production of indoleamine 2,3-dioxigenase 1 (IDO1), an anti-inflammatory enzyme, can be stimulated by the IFN-γ released by NK cells after TIM-3/Gal-9 binding [146]. Goncalves Silva et al. reported that the soluble form of TIM-3, formed by its shedding from the surface of AML blasts, inhibits the release of interleukin2 (IL-2), involved in the activation and function of T- and NK cells [147]. Its high expression on immune cells has been associated with a worse prognosis in solid and hematologic neoplasms [105,130]. Li et al. found higher TIM-3 overexpression in CD4+ T-cells from patients with FLT3-mutated compared to non-mutated AML, and in CD8+ cells of high-risk compared to low-risk AML patients [148]. Kong et al. reported shorter leukemia-free survival (LFS) after HCT in patients with high numbers of TIM-3/PD-1 co-expressing T-cells [126]. Zahran et al. demonstrated the association between TIM-3 upregulation and poor prognosis in AML with normal cytogenetics [149]. Tan et al. reported that the overexpression of TIM-3 on CD8+ cells was associated with a negative prognosis [150]. Less defined is the prognostic role of TIM-3 expression in leukemic cells: Tan et al. observed the highest TIM-3 level in M4 AMLs [151]. Dama et al. correlated TIM-3 expression on blasts and chemotherapy failure [152], while Xu et al. reported a better response to chemotherapy in cases with TIM-3-expressing blasts [153]. More studies are needed to clarify the relationship between the expression of TIM-3 on AML blast cells and response to chemotherapy and, which is most important, the mechanisms inducing the up-regulation of TIM-3 on LCSs and its possible impact on the biology of LSCs.

However, taken together, the available data points to TIM-3 as an ideal candidate for therapy with monoclonal antibodies (MoAb). The inhibition of TIM-3/gal-9 binding in “vitro” reduced the proliferation capacity of AML cells [154], and, in murine models, an anti-human TIM-3 MoAb eliminated LSCs without affecting normal hematopoiesis [145]. At present, several MoAbs are being tested in trials for solid tumors, such as MBG453 (sabatolimab), TSR-022, BMS-986258, LY3321367, SYM023, BGB-A425, SHR 1702 [155], but only MBG453 has shown preliminary efficacy and safety in AML and MDS. Currently, eight phase I/II trials of MBG453 in monotherapy or in combination with HMAs, PD-1 inhibitors, the MDM2 inhibitor HDM201, and venetoclax are ongoing. The interim data from the NCT03066648 trial (MBG453 + HMAs) reported an ORR of 58% in MDS and 38% in newly diagnosed AML patients for the MBG453 + DAC arm and 70% in MDS and 27% in AML patients for the MBG453 + AZA arm, respectively. The commonest grade 3/4 AEs were thrombocytopenia, anemia and neutropenia (febrile or not); in the MBG453 + DAC group, four immune-related events were reported (ALT increase, arthritis, hepatitis and hypothyroidism), compared to none in the MBG453 + AZA cohort [156,157].

### 4.4. The LAG-3/MHC Axis

LAG-3 (CD223) is a CD4-like molecule binding MHC class II with a greater affinity than CD4, generating a signal blocking T-cell activation [115]. The protein is coded by a gene in the short arm of chromosome 12 (12p13.32) in humans and in chromosome 6 in mice. The gene encodes a type I membrane protein with a molecular weight of 70kDa [158]. This locus is adjacent and has a similar intron/exon organization to that of the CD4 co-receptor, suggesting that *lag-3* and *CD4* genes evolved via gene duplication from a pre-existing common ancestor gene. The mature protein consists of four extracellular immunoglobulin-like domains, a TM region containing a cleavage site mediated by two metalloproteases (ADAM 10 and ADAM 17) induced by TCR signaling, and a cytoplasmatic tail that mediates the intracellular transduction of inhibitory signals [158,159,160]. In addition, the LAG-3 cytoplasmic domain influences the protein membrane location. After TCR engagement LAG-3 co-localizes with CD3, CD4 or CD8 in the immunological synapses [161]. Apart from the cell membrane, LAG-3 is stored in lysosomes and can be translocated to the membrane after T-cell activation [162]. MHC-II molecules are considered canonical LAG-3 ligands and are in competition with CD4 binding. The engagement with its ligand reduces T-cell activity and cytokine secretion, blocking T-cell activation and function [115]. In addition, LAG-3 can bind Galectin-3 (Gal-3), a lectin expressed in various tumors and in activated T-cells. In tumors, Gal-3/LAG-3 binding inhibits anti-tumor-specific immune response by suppressing CD8 cytotoxic function and via the inhibition of plasmacytoid dendritic cell expansion [163]. Recently the fibrinogen-like protein1 (FGL1), a member of the fibrinogen family, has been identified as another LAG-3 ligand and has been proposed as an alternative mechanism of immune evasion. FGL1 expression is induced by IL-6, and it is highly upregulated in many solid tumors and is associated with poor prognosis and resistance to PD1 therapy [164]. The molecule is expressed on activated CD4+ and CD8+ T-cells, Tregs, NK cells, B-cells and dendritic cells [130]. On CD4+ cells, LAG-3 is present in Th0 and Th1 but not by Th2 [158]. Like other checkpoint molecules, LAG-3 has been identified in Tregs in the cancer microenvironment, in both natural and inducible activated subsets [165], where the IL-27/LAG-3 axis enhances the suppressive function [166]. The frequent co-expression with PD1 suggests a comparable function to PD1 [116,131]. In tumors, LAG-3 is expressed by exhausted CD8+/PD-1+ tumor-infiltrating lymphocytes [167], but its role in immune escape is still controversial. In B-cells, LAG-3 expression is induced by T-cells [168]. In addition, the molecule has been found in a subset of natural regulatory plasma cells (LAG-3/CD138 high) [169] and chronic lymphocytic leukemia [170]. 

In solid cancers, LAG-3 and PD-1 co-expression has been associated with poor sensitivity to PD-1 blockade and has been proposed as a biomarker for predicting the efficacy of immunotherapy, while data relating to AML are still scarce [129,171,172]. It should be remembered that the expression of MHC class II in AML blasts can be involved in both immune suppression and antigen presentation processes. Antibodies targeting LAG-3 are currently being tested in solid tumors, lymphomas and multiple myeloma, either in monotherapy or in combination with PD-1 inhibitors [115,130], with encouraging results, especially in metastatic melanoma, where the combination nivolumab + relatlimab significantly improved PFS. At present, only one clinical trial has been activated in AML patients: the AARON study (NCT04913922), which will test the safety and tolerability of a triplet of AZA, nivolumab (anti-PD-1) and relatlimab (anti-LAG-3) in patients with relapsed/refractory AML and newly diagnosed AML aged > 65 years: recruitment started in November 2022; no results are still available. 

### 4.5. The CD200/CD200R Axis

CD200 is a highly conserved 48 kDa type-1 trans membrane glycoprotein, structurally related to the B7 family, encoded on chromosome 3q12-q13, which is close to the region coding for CD80/CD86 proteins. Through interaction with its receptor CD200R, CD200 causes the weakening of many immune-responsive effects, resulting in the prolonged survival of transplanted allograph, but also in tolerance toward tumor cells. CD200 overexpression has been reported both in solid tumors and in AML, where it marks LSCs but not the normal stem cell counterpart [130,173]. CD200 overexpression in AML has been associated with worse outcomes, mainly in the favorable prognostic groups [174,175]. CD200 expression on leukemic cells suppresses memory T-cell function, expands Tregs, and downmodulates NK function [176,177,178]. “In vitro” and “in vivo” murine models clearly demonstrated that the inhibition of the CD200/CD200R axis with MoAbs restores anti-AML immune response [179]. At present, the anti-CD200 MoAb samalizumab is being tested in solid tumors. In hematological malignancies, a phase I trial in CLL and multiple myeloma (NCT00648739) has been prematurely terminated for administrative reasons, not for safety concerns. At present, the only active trial of samalizumab in AML (the Beat AML trial, NCT03013998) has completed the recruitment. The aim of the study was the identification of biomarkers to drive the assignment of AML patients to the most appropriate investigational study; no results are yet available.

### 4.6. The CD27/CD70 Axis

Among costimulatory pathways, CD27/CD70 has gained interest after demonstrating that it acts as a switch between immunity and tolerance. CD27, a member of the tumor necrosis factor superfamily, acts as a potent costimulatory molecule, providing a second signal for T-cell activation [180]. Different from other costimulatory molecules, it is constitutively expressed on resting naïve and memory T-cells and in a subset of NK cells [180], as well on memory B-cells and in most B-cell lymphomas [181]. CD27 signal is controlled by its unique ligand CD70, which is transiently upregulated on immune cells upon activation, but that is not expressed in normal tissues and hematopoietic system during homeostasis, suggesting that the early hematopoietic stages are independent of the CD27/CD70 axis [182]. There is evidence that CD27 engagement may produce different results depending on the strength, duration, and timing of the stimulation. After a short triggering, CD27/CD70 interaction provides a second signal for the differentiation of naïve CD4+ T-cells into Th1 effectors and of CD8+ T-cells into cytotoxic lymphocytes [183,184,185]. Conversely, the chronic or persistent triggering of CD27 leads to T-cell exhaustion and activation-induced cell death [186]. Moreover, chronic CD27/CD70 engagement maintains DCs in a tolerogenic state and favors the expansion of peripheral Tregs, suppressing anti-tumor immune responses [187]. Upon activation by CD70, the extracellular domain of CD27 is cleaved and is found in a soluble fragment (sCD27) in body fluids [188]. In tumors, an aberrant CD70 expression has been found without (solid tumors) or with (hematologic malignancies) CD27 co-expression, facilitating immune escape and contributing to generating a BM-suppressive microenvironment [188]. In LSCs, aberrant CD27 expression was detected in AML and in chronic myeloid leukemia (CML), and high levels of sCD27 were associated with a poor prognosis in terms of AML [189]. In LSCs, the CD27/CD70 pathway induces the aberrant activation of the Wnt pathway, which promotes LSC proliferation, drug resistance and disease progression [189,190]. Furthermore, the CD27/CD70 pathway can affect blast survival by regulating the MEK pathway, transcription factor AP-1 and the Wnt pathway through the activation of β-catenin [189]. With these premises, targeting the CD27/CD70 axis may constitute an effective immunotherapeutic strategy. However, the activity of anti-CD70 antibody–drug conjugates relies on the internalization of the drug, which can significantly differ among tumor cells [191]. More promising are antibodies that are able to induce cellular cytotoxicity (ADCC). The anti-CD79 antibody cusatuzumab (ARGX-110) showed the ability to elicit both ACC and ADCC because of afucosylation that enhances the binding to FcγRIIIa in a phase I trial in advanced malignancies [192]. Xenograft transplantation models and in vitro experiments proved cusatuzumab activity in eliminating LSCs by inducing differentiation and increasing apoptosis [193]. Based on the preclinical data, a phase I/II trial assessing the safety and efficacy of cusatuzumab as monotherapy and in combination with AZA in untreated AML patients unfit for intensive therapy has been designed (NCT0030612). The best hematological response was CR/CRi (8/2 patients), and the median time to response was 3.9 months, and a durable response was observed in six patients, with median PFS not being reached. At present, another trial evaluating cusatuzumab and AZA in newly diagnosed AML or high-risk MDSs unfit for intensive therapy has been completed (NCT042415499), but the results are not yet available. Lastly, two clinical studies for newly diagnosed AMLs unfit for intensive chemotherapy evaluating the combination of cusatuzumab + AZA (CULMINATE trial—NCT04023526) and cusatuzumab + AZA + venetoclax (ELEVATE trial—NCT04150887) are active but not recruiting.

### 4.7. The TIGIT-CD155/CD112 Axis

TIGIT is a protein coded by a gene located on chromosome 3q13.31; it includes an extracellular IgV region, a TM domain and a cytoplasmic tail that harbors a canonical ITIM and a tail-tyrosine (ITT)-like phosphorylation motif. Its expression is restricted to T-cells and NK cells [194]. TIGIT binds to ligands CD155 and CD122 with different affinities and exerts its immunosuppressive function by delivering inhibitory signals in competition with CD226 and CD96 counterparts, providing costimulatory signals [195]. It was first identified by Yu et al. in 2009 as an immune checkpoint that suppresses T-cell activation by promoting the generation of mature immunoregulatory DCs [194]. Beyond its indirect inhibitory function mediated by DCs, TIGIT engagement also produces a direct suppressive activity on T-cells and NK cells [196,197]. CD155 is expressed at low levels in many normal tissues, but the upregulation of CD155 is reported in many tumor cells, such as melanoma, pancreatic cancer, lung adenocarcinoma, colon cancer, and glioblastoma, and it is involved in tumor development and invasion and is correlated with poor clinical outcomes [198].

The role of TIGIT in AML is less defined. Kong et al. reported that TIGIT expression on peripheral T-cells from AML patients correlated with CD8+ T-cell exhaustion and poor prognosis [199]. Liu et at. found a correlation between TIGIT expression and a dysfunctional NK phenotype in AML [200]. In addition, TIGIT was often co-expressed with other inhibitory receptors, such as PD-1, TIM-3 and LAG- 3, and this was associated with the reduced functionality of CD8+ cytotoxic T-cells [201]. Goumay et al. linked TIGIT expression on peripheral CD4+ T-cells after HCT with disease recurrence [202]. Moreover, the high expression of the TIGIT ligand on leukemic cells may be a mechanism of immune evasion [203], and “in vitro” studies in AML cell lines or in primary AML cells demonstrated that blocking the TIGIT-CD155/CD112 axis may improve anti-leukemia immune response [198]. Consequently, an antibody-mediated blockade of the TIGIT-CD155/CD112 pathway could represent a promising approach. Many clinical trials testing anti-TIGIT antibodies in solid tumors are ongoing, but until now, there have been no active trials in relation to AML. 

### 4.8. The CD47/SIRPs Axis

CD47 (formerly known as integrin-associated protein) is a TM protein encoded on chromosome 3q13.12 that functions as an anti-phagocytic or “do not eat me” signal, enabling the production of CD47-expressing cells to elude macrophages. The inhibition of phagocytosis depends on CD47 binding to its cognate receptor, the signal regulatory protein alpha (SIRPα), on macrophages and DCs, which initiates a signal transduction cascade that activates tyrosine phosphatase, leading to the inhibition of myosin accumulation at the phagocytic synapse, thus preventing phagocytosis [204]. CD47 was first found in red cells, but current evidence indicates that it is widely expressed in many normal cells. In oncology, CD47 was initially discovered in human ovarian cancer in the late 1980s [205]. Since then, its expression has been found in many solid tumors and hematologic malignancies [206,207] and, at present, it is considered to be a universal mechanism of immune evasion by cancer cells.

The clinical relevance of CD47 expression has been extensively evaluated in relation to AML patients, where high CD47 levels on the cell surface and high CD47 mRNA levels were associated with a poor prognosis [206]. Moreover, in AML, CD47 expression was increased in CD34+CD38-CD90-lin- LSCs but not in their normal counterpart, making the molecule a potential target for anti-leukemic therapies. “In vitro” data with anti-CD47 antibodies confirmed the selective reduction in LSCs; in mice models, the failure of leukemia transplant in secondary recipients demonstrated LSC depletion/dysfunction [208]. As previously suggested, it must be underlined that CD47 is also widely expressed in normal cells, and this might increase off-target toxicity when CD47 is considered to be a therapeutic target in leukemia. In fact, “in vitro” studies comparing normal and leukemic cells showed that both evade phagocytosis by expressing CD47. However, only cancer cells also expressed pro-phagocytic signals, so that after blocking CD47, the “eat me signals” were activated only in cancer cells. Chao et al. identified calreticulin as the “dominant” pro-apoptotic signal in cancer cells and demonstrated that calreticulin completely blocks abrogated, in vitro, anti-CD47 mediated cancer cell phagocytosis [209]. In addition, blocking the CD47/SIRPα axis can induce M1 polarization in tumor-associated macrophages and increase macrophage recruitment [210], and can promote phagocytosis by DSs and consequent antigen presentation to CD8+ T-cells, inducing an adaptive anti-tumor immune response [208]. CD47 antagonists can also enhance NK-mediated antibody-dependent cytotoxicity (ADCC) and complement-mediated cytotoxicity, promote cancer cell apoptosis, reduce cancer cell proliferation, and prevent tumor cell migration [211,212,213].

According to these pieces of evidence, anti-CD47 antibodies were developed for clinical use. CC-90002 was the first generation of humanized anti-CD47 MoAb to enter clinical research. Although preclinical studies have demonstrated its potential efficacy, a phase I study of patients with relapsed/refractory AML and MDS (NCT02641002) was terminated due to a lack of efficacy in monotherapy, as none of the 28 patients enrolled showed significant benefit from CC-9002 monotherapy [214]. The reason for efficacy failure was attributed to the switch from IgG1 and IgG4 that significantly weakened its killing ability; at present, the research in hematological malignancies was stopped. In 2020, the FDA granted breakthrough therapy designation to another anti-CD47 antibody, magrolimab, following evidence of the activity in terms of MDS. In high-risk MDS, a phase 1b trial of magrolimab and AZA proved a favorable safety profile, as the most common treatment-emergent adverse events (TEAEs) were constipation (68%), thrombocytopenia (55%), anemia (52%), neutropenia (47%), nausea (46%), diarrhea (44%). The responses to therapy were encouraging, with CR and ORR rates of 33% and 76%, respectively, and an OS probability at 12 and 24 months of 75% and 52%, respectively [215]. Favorable outcomes were observed even in patients with TP53 mutation (40%). Similar results came from a phase 1b trial of frontline magrolimab +AZA in patients with TP53-mutated AML: the ORR was 48% (CR 33.3% and CRi 8.3%), 30- and 60 days mortality were 8.3% and 18.1%, respectively, and the median OS was 10.8 months. The TEAE profile was superimposable on a previous trial [216]. A phase 3 trial of magrolimab + AZA vs. standard of care in TP53 mutated AML is ongoing (ENHANCE2, NCT04778397), and a phase 3 randomized trial of AZA + magrolimab or placebo in MDS is recruiting (ENHANCE trial—NCT04313881). Another phase 2 clinical trial of magrolimab + AZA + venetoclax vs. magrolimab + intensive chemotherapy (mitoxantrone, etoposide, and cytarabine) vs. magrolimab + oral AZA is open and recruiting (NCT04778410).

It is worth remembering that in 2022, the FDA suspended trials evaluating the association of magrolimab + AZA due to discrepancies in the investigators reporting severe adverse reactions; the trials were restarted after a revision of the safety data, but many studies were negatively impacted by the action. Currently, other different combinations, including magrolimab, are under investigation in MDS/AML: magrolimab + anti-PD-L1 atezolizumab (NCT03922477), magrolimab + AZA + venetoclax (NCT0443691), magrolimab + daratumumab, magrolimab + pomalidomide and dexamethasone, magrolimab + bortezomid and dexamethasone (NCT04892446).

Evorpacept (ALX148) is a high-affinity CD47-blocking protein with a modified Fc domain to circumvent red cell agglutination, reported with magrolimab. There are two ongoing trials assessing ALX148 combined with AZA in high-risk MDS (ASPEN02 trial—NCT04417517) and with AZA + venetoclax in AML (ASPEN05 trial—NCT04755244). The preliminary results of the phase 1 part of the ASPEN05 trial indicate the tolerability of the combination evorpacept + AZA [217]. TEAEs were generally mild, and there were no evorpacept-related severe AEs. Anti-leukemic activity was observed in subjects with both newly diagnosed and relapsed/refractory AML (including patients previously treated with venetoclax), with BM blast reduction ranging from 20 to 100%. The combination will be further evaluated in the randomized phase 2 part of the study.

Lemzoparlimab (TJC4, TJ011133)) is a second-generation anti-CD47 IgG4 antibody with a unique binding epitope and red-cell-sparing properties developed in China. A phase II clinical trial in relapsed/refractory AML and MDS is currently ongoing (NCT04202003). Preliminary results show grade 3 AEs in only one out of five of the enrolled patients, and one patient attained morphologically defined leukemia-free status [218]. Lemzoparlimab, in combination with AZA, has been approved for a phase III clinical trial for high-risk MDS.

The other five anti-CD47 antibodies are, at present, under investigation in AML, either as monotherapy or in association with AZA. Clinical data on anti-CD47 are still incomplete, but the burst of new molecules targeting the CD47/SIRPα axis, including bispecific antibodies and SIRPα/Fc fusion protein antibodies, is likely to significantly change the therapeutic landscape in terms of AML.

## 5. Conclusions

In the past ten years, technological progress and better knowledge of the genetic basis of leukemogenesis have significantly increased the therapeutic tools used for AML. The evidence of an immune-suppressive BM environment and the availability of molecules that are potentially able to restore immune response and elicit anti-leukemia immune surveillance open new perspectives. The advent of checkpoint inhibitors in solid cancer has led to relevant improvement, mostly in tumors with limited therapeutic options. In AML, despite solid biological bases, the results appear less remarkable; a recent meta-analysis on ICIs in AML confirms the modest results reported by trials [219]. As expected, ORR, CR and OS were better in the first-line setting than in relapsed patients, but it must be underlined that virtually all studies employed ICIs in association with different conventional therapies, and the actual role of ICIs is not evaluable. Moreover, it should be kept in mind that in multi-relapsed/refractory patients, immune cell exhaustion is probably increased. Finally, it should be remembered that under oncologic conditions, AML has one of the lowest mutational burdens and that neoantigen formation has a significant impact on immunotherapy efficacy. The identification of biomarkers predicting the efficacy and design of novel prospective trials that stratify patients according to different biological profiles will hopefully contribute to improving the clinical results and may help to properly allocate these new therapies. 

## Figures and Tables

**Figure 1 biomedicines-11-01724-f001:**
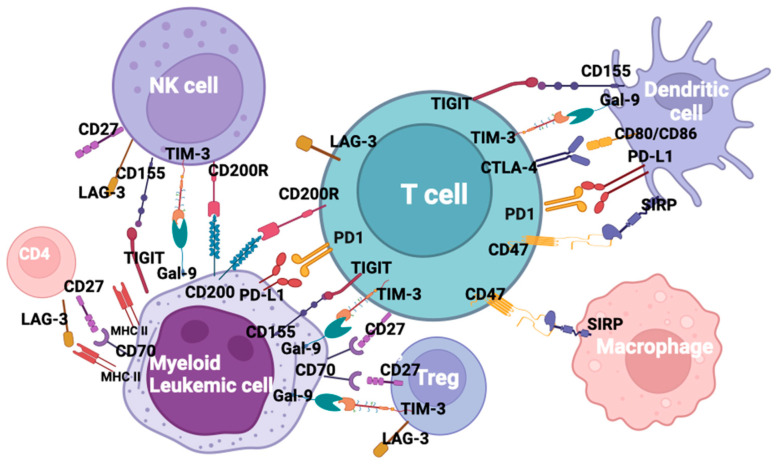
Checkpoint inhibitors and ligands in acute myeloid leukemia. Created in Biorender.com.

## Data Availability

No new data were created or analyzed in this study. Data sharing is not applicable to this article.

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
