# Peer review of "Checkpoint Inhibitors in Acute Myeloid Leukemia"

_biomedicines, 2023, doi:10.3390/biomedicines11061724_

Round 1

Reviewer 1 Report

This is a well written and comprehensive review, evaluating the basis of immune dysfunction in AML as well as the various immune checkpoint inhibitor trials.

This is a difficult topic, as several papers, sometimes conflicting, have been published on the topic, and credit should be given to the authors as they approached such a vast topic.

The main issue noted was the citing of reviews articles (in some areas of the review) rather than of the original research work. While the authors have comprehensively reviewed the existing literature in this area, a few references need to be added in pertinent sections, as further detailed below. Additional suggestions are also mentioned:

1. Line 25-30

Authors state that it is well-accepted that malignant hematopoiesis mirror its normal counterpart, with "stem-like" progenitors giving rise to partially committed progenitors etc.

This actually remains an area of controversy and I would probably eliminate this paragraph entirely, as it is not central to the focus of this review.

2. Line 34.

Please remove reference 3 and cite original work (TCGA paper, Papaemmanuil paper, and Beat AML paper)

3. Line 79 and 80

It is well known that patients receiving T-cell depleted allogeneic HCT have  higher risk of graft failure [14]. However, when CD3 cells were co-administered with allogeneic BM-derived stem cell a full engraftment can be obtained [15]

This implies that the T cells support engraftment alone whereas its effect is more nuanced given the role it plays  generating a graft versus leukemia effect which is well established in correlating with prognosis. I would phrase this more carefully.

4. The following citation should be added and explained in the main text

For AML immune dysfunction

- A recent study by Ferraro F et al (10.1073/pnas.2116427118) has shown that immunological phenotypes of AML patients at diagnosis are an important determinant of outcome to standard of care therapies. This study highlights that patient with better outcome to standard of care therapies have more CD4+ T cells in their BM, intact response to T cell stimulation and "cold" infiltrates. While patient who relapse havehigher expression of immunomodulatory molecules on AML cells (MHC class II, CD200 and MRC1) as well as exhaustion markers on T cells (LAG, CD200R, CD38, TIM3). This paper also show how bloking the LAG3-MHC class II axis can restore in-vitro activation of 60-70% of the AML cells (this can also be added to the LAG3 section of the review)

- Another study worth mentioning is the study by Lamble at al (10.1073/pnas.1916206117) which also highlights the association between T cell infiltration of AML at diagnosis and outcome.

- A beautiful original work first proposing the association between immunosuppression and outcome after chemotherapy in AML was published in 1971 in the NEJM:  E. M. Hersh, J. P. Whitecar Jr., K. B. McCredie, G. P. Bodey Sr., E. J. Freireich, Chemotherapy, immunocompetence, immunosuppression and prognosis in acute leukemia. N. Engl. J. Med. 285, 1211–1216 (1971)

Role of mutations in shaping the immune environment

Please add the following references

- O. Dufva et al., Immunogenomic landscape of hematological malignancies. Cancer Cell 38, 380–399.e13 (2020).

- P. S. Chauhan et al., Mutation of NPM1 and FLT3 genes in acute myeloid leukemia and their association with clinical and immunophenotypic features.

- A recent Nature Communication paper also established some of the mechanism behind how somatic mutations in AML cells shape the immune environment: (10.1038/s41467-023-37592-9)

Immune escape via downregulation of HLA class II

The authors have mentioned the downregulation of HLA class II as well as the role of epigenetic mechansims in mediating this effect. However, two very important studies in this field which proved this on large datasets of patients are the one by Christopher et al (N Engl J Med 2018; 379:2330-2341) and Toffalori C et al (Nat Med. 2019 Apr;25(4):603-611).

5. Paragraph from line 121-135: only 2 reviews are cited here. Please add the references of the original research.

Minor comments/edits:

a.Line 536

CD27/CD70 – the word axis to be included

b. line 517

CD200/CD200. The letter R in the CD200R ligand is omitted.

c.line 406

% is repeated twice

d.line 343

on the other hands is written instead of on the other hand

e.line 182

and is associate should read and is associated.

Appropriate, only needing minor proofreading for few typos.

Author Response

This is a well written and comprehensive review, evaluating the basis of immune dysfunction in AML as well as the various immune checkpoint inhibitor trials.

This is a difficult topic, as several papers, sometimes conflicting, have been published on the topic, and credit should be given to the authors as they approached such a vast topic.

We thank the reviewer for his/her appreciation of our work.

The main issue noted was the citing of reviews articles (in some areas of the review) rather than of the original research work. While the authors have comprehensively reviewed the existing literature in this area, a few references need to be added in pertinent sections, as further detailed below. Additional suggestions are also mentioned:

  1. Line 25-30

Authors state that it is well-accepted that malignant hematopoiesis mirrors its normal counterpart, with "stem-like" progenitors giving rise to partially committed progenitors etc.

This actually remains an area of controversy and I would probably eliminate this paragraph entirely, as it is not central to the focus of this review.

We agree with the reviewer; the paragraph has been almost entirely eliminated.

  1. Line 34.

Please remove reference 3 and cite original work (TCGA paper, Papaemmanuil paper, and Beat AML paper)

Reference has been changed with the original works.

  1. Line 79 and 80

It is well known that patients receiving T-cell depleted allogeneic HCT have higher risk of graft failure [14]. However, when CD3 cells were co-administered with allogeneic BM-derived stem cell a full engraftment can be obtained [15]

This implies that the T cells support engraftment alone whereas its effect is more nuanced given the role it plays generating a graft versus leukemia effect which is well established in correlating with prognosis. I would phrase this more carefully.

The sentence has been rephrased.

  1. The following citation should be added and explained in the main text

For AML immune dysfunction

- A recent study by Ferraro F et al (10.1073/pnas.2116427118) has shown that immunological phenotypes of AML patients at diagnosis are an important determinant of outcome to standard of care therapies. This study highlights that patient with better outcome to standard of care therapies have more CD4+ T cells in their BM, intact response to T cell stimulation and "cold" infiltrates. While patient who relapse have higher expression of immunomodulatory molecules on AML cells (MHC class II, CD200 and MRC1) as well as exhaustion markers on T cells (LAG, CD200R, CD38, TIM3). This paper also shows how blocking the LAG3-MHC class II axis can restore in-vitro activation of 60-70% of the AML cells (this can also be added to the LAG3 section of the review)

- Another study worth mentioning is the study by Lamble and al. (10.1073/pnas.1916206117) which also highlights the association between T cell infiltration of AML at diagnosis and outcome.

- A beautiful original work first proposing the association between immunosuppression and outcome after chemotherapy in AML was published in 1971 in the NEJM:  E. M. Hersh, J. P. Whitecar Jr., K. B. McCredie, G. P. Bodey Sr., E. J. Freireich, Chemotherapy, immunocompetence, immunosuppression and prognosis in acute leukemia. N. Engl. J. Med. 285, 1211–1216 (1971)

The suggested papers have been included.

Role of mutations in shaping the immune environment

Please add the following references

- O. Dufva et al., Immunogenomic landscape of hematological malignancies. Cancer Cell 38, 380–399.e13 (2020).

- P. S. Chauhan et al., Mutation of NPM1 and FLT3 genes in acute myeloid leukemia and their association with clinical and immunophenotypic features.

- A recent Nature Communication paper also established some of the mechanism behind how somatic mutations in AML cells shape the immune environment: (10.1038/s41467-023-37592-9)

The suggested references have been added.

Immune escape via downregulation of HLA class II

The authors have mentioned the downregulation of HLA class II as well as the role of epigenetic mechanisms in mediating this effect. However, two very important studies in this field which proved this on large datasets of patients are the one by Christopher et al (N Engl J Med 2018; 379:2330-2341) and Toffalori C et al (Nat Med. 2019 Apr;25(4):603-611).

The suggested references have been added in the paragraph.

  1. Paragraph from line 121-135: only 2 reviews are cited here. Please add the references of the original research.

References of the original research have been added in the first part of the paragraph (differences between low-risk MDS and high-risk MDS /AML. For the second part of the paragraph, for sake of brevity and clarity, the cited review (Austin R et al, CROH 2016) has been maintained.  

Minor comments/edits:

  1. line 536

CD27/CD70 – the word axis to be included.

Corrected.

  1. line 517

CD200/CD200. The letter R in the CD200R ligand is omitted.

Corrected.

  1. line 406

% is repeated twice

Corrected.

  1. line 343

on the other hands is written instead of on the other hand

Corrected.

  1. line 182

and is associate should read and is associated.

Corrected.

Reviewer 2 Report

This review summarized various components in the microenvironment of acute myeloid leukemia (AML) and also showed immune checkpoint inhibitors (ICIs), such as monoclonal antibodies, in combination with other therapeutics in AML. However, the manuscript lacks high-quality figures to show the relationship between different factors in the microenvironment and also indicate the roles of ICIs as therapeutics. With only text descriptions, readers feel burdensome to summarize, compare, and effectively analyze various intrinsic and extrinsic factors. As such, we cannot easily obtain a good landscape view of your paper. This issue be seen in the introduction of current treatments (in the introduction section), illustration of T cells in the bone marrow microenvironment (in section 2, line 62), and discussion of mutated genes (line 163-183). Using tables or schematic illustrations can not only make the review more attractive but also reduce the overwhelming text redundancy. For example, starting from line 349, the authors discussed various antibody inhibitors in clinical trials; these can be compiled into a table for a clear illustration to readers. In addition, the authors summarized lots of therapeutic antibodies in the text. However, readers are eager to obtain insights from such a summary. From my point of view, the discussion in this paper is not deep enough to show readers fruitful perspectives or insights. Lastly, authors need to simplify the language and check the spelling issues. For instance, in line 140, the word “moleculae” is misspelled. Overall, the review makes some efforts in reviewing AML and ICIs, although figures, tables, and insights are not satisfactory for our readers.  

English is fine. 

Author Response

Reviewer 2

This review summarized various components in the microenvironment of acute myeloid leukemia (AML) and also showed immune checkpoint inhibitors (ICIs), such as monoclonal antibodies, in combination with other therapeutics in AML. However, the manuscript lacks high-quality figures to show the relationship between different factors in the microenvironment and also indicate the roles of ICIs as therapeutics. With only text descriptions, readers feel burdensome to summarize, compare, and effectively analyze various intrinsic and extrinsic factors. As such, we cannot easily obtain a good landscape view of your paper. This issue be seen in the introduction of current treatments (in the introduction section), illustration of T cells in the bone marrow microenvironment (in section 2, line 62), and discussion of mutated genes (line 163-183). Using tables or schematic illustrations can not only make the review more attractive but also reduce the overwhelming text redundancy. For example, starting from line 349, the authors discussed various antibody inhibitors in clinical trials; these can be compiled into a table for a clear illustration to readers. In addition, the authors summarized lots of therapeutic antibodies in the text. However, readers are eager to obtain insights from such a summary. From my point of view, the discussion in this paper is not deep enough to show readers fruitful perspectives or insights. Lastly, authors need to simplify the language and check the spelling issues. For instance, in line 140, the word “moleculae” is misspelled. Overall, the review makes some efforts in reviewing AML and ICIs, although figures, tables, and insights are not satisfactory for our readers.  

We agree with the reviewer that, due to the vastity of the topic and the number of recent evidence published, the manuscript could be sometimes redundant; we tried to simplify most parts of the paper and insert a table to substitute and/or integrate the text about clinical trials on drugs targeting the PD1/PD-L1 axis (paragraph 4.2).

More, the manuscript has been thoroughly revised for typos, according also to comments of reviewer 1.